# In Situ Fabrication of SnS_2_/SnO_2_ Heterostructures for Boosting Formaldehyde−Sensing Properties at Room Temperature

**DOI:** 10.3390/nano13172493

**Published:** 2023-09-04

**Authors:** Dan Meng, Zongsheng Xie, Mingyue Wang, Juhua Xu, Xiaoguang San, Jian Qi, Yue Zhang, Guosheng Wang, Quan Jin

**Affiliations:** 1College of Chemical Engineering, Shenyang University of Chemical Technology, Shenyang 110142, China; mengdan0610@hotmail.com (D.M.); xzs19991015@163.com (Z.X.); meat_d@163.com (Y.Z.); m18240179987@163.com (G.W.); 2Institute for Superconducting and Electronic Materials (ISEM), Australian Institute for Innovative Materials (AIIM), Innovation Campus, University of Wollongong, Squires Way, North Wollongong, NSW 2500, Australia; mw663@uowmail.edu.au; 3Key Laboratory of Automobile Materials (Ministry of Education), School of Materials Science and Engineering, Jilin University, Changchun 130022, China; hmm9911057@163.com; 4State Key Laboratory of Biochemical Engineering, Institute of Process Engineering, Chinese Academy of Sciences, Beijing 100190, China; jqi@ipe.ac.cn

**Keywords:** SnS_2_/SnO_2_, n–n heterostructures, formaldehyde sensing, room temperature, DFT calculations

## Abstract

Formaldehyde, as a harmful gas produced by materials used for decorative purposes, has a serious impact on human health, and is also the focus and difficulty of indoor environmental polution prevention; hence, designing and developing gas sensors for the selective measurement of formaldehyde at room temperature is an urgent task. Herein, a series of SnS_2_/SnO_2_ composites with hollow spherical structures were prepared by a facile hydrothermal approach for the purpose of formaldehyde sensing at room temperature. These novel hierarchical structured SnS_2_/SnO_2_ composites−based gas sensors demonstrate remarkable selectivity towards formaldehyde within the concentration range of sub-ppm (0.1 ppm) to ppm (10 ppm) at room temperature. Notably, the SnS_2_/SnO_2_−2 sensor exhibits an exceptional formaldehyde-sensing performance, featuring an ultra-high response (1.93, 0.1 ppm and 17.51, 10 ppm), as well as good repeatability, long-term stability, and an outstanding theoretical detection limit. The superior sensing capabilities of the SnS_2_/SnO_2_ composites can be attributed to multiple factors, including enhanced formaldehyde adsorption, larger specific surface area and porosity of the hollow structure, as well as the synergistic interfacial incorporation of the SnS_2_/SnO_2_ heterojunction. Overall, the excellent gas sensing performance of SnS_2_/SnO_2_ hollow spheres has opened up a new way for their detection of trace formaldehyde at room temperature.

## 1. Introduction

Formaldehyde in indoor environments mainly comes from plywood, density board, paint, etc., in decoration materials. Due to the large use of urea formaldehyde resin adhesive materials, formaldehyde pollution in indoor air is aggravated [1], which causes serious health concerns. Even low concentrations of formaldehyde can cause breathing difficulties and headaches in humans [2]. More importantly, high concentrations of formaldehyde can disrupt the central nervous system and immune system and cause respiratory diseases and blindness [3]. Notably, multiple medical organizations and institutions around the world have proposed that formaldehyde’s maximum permissible occupational exposure concentration is 0.75 ppm over 8 h and 2 ppm over 15 min [4]; therefore, accurate detection of trance formaldehyde is highly desirable for human health.

Currently, many analytical technologies, e.g., electrochemistry [5], chromatography [6], fluorescence [7], and spectrophotometry [8], are available for detecting formaldehyde; however, they need to overcome the drawbacks of large equipment size and expensive and complicated operation, which limit their application in daily life. Under these circumstances, metal oxide semiconductors (MOS) have attracted great attention as gas sensors because of their simple operation, easy fabrication, and real-time monitoring [9,10], which have become a very effective means of formaldehyde gas detection. In the past five years, different types of MOS materials, such as ZnO [11], SnO_2_ [12], NiO [13], In_2_O_3_ [14], and CuO [15], have been used to fabricate gas sensors and offer good sensitivity to formaldehyde. Among them, n-type tin dioxide (SnO_2_) has attracted widespread attention for monitoring various reduced or oxide gases due to its excellent thermal stability [16], compatibility, and multiple morphologies [17,18]. Research has shown that SnO_2_ could be a promising formaldehyde gas-sensing material. Nonetheless, the commonly used SnO_2_ sensor shows low sensitivity and poor selectivity and usually works under high-power consumption limited to fulfilling the demands of integration and intelligence; therefore, room-temperature operating sensing materials with the capability to detect formaldehyde are being intensively explored to meet the requirement for portable or Internet of Things applications [19].

In recent years, many published studies have demonstrated that the three-dimensional (3D) hierarchical micro-nano structures of SnO_2_ possess numerous active sites and abundant mesopores. These structural features facilitate rapid gas diffusion and adsorption of gas molecules on the surface, thereby imparting excellent gas sensing properties [20,21,22]; however, characterizing the gas-sensing capabilities of these structures at room temperature presents a significant challenge. Recently, researchers have made considerable efforts to address this issue by constructing heterostructures using one or more different materials. These heterojunctions have been shown to have a noticeable impact on adjusting the intrinsic electronic properties of SnO_2_, thereby achieving an outstanding gas-sensing performance [23,24,25,26]. Particularly, tin disulfide (SnS_2_), as a representative Sn-based materials, has attracted much attention from researchers as an important platform for heterostructure-sensing applications owing to its layered structure, appropriate mid-bandgap, high surface activity, easy surface functionalization, and large electronegativity [27]. The SnO_2_ band is more positive than the SnS_2_ band, resulting in the photoexcitation of electrons and their movement from the SnS_2_ conduction band to SnO_2_ until the equilibrium of the Fermi level is reached. This process forms n–n heterojunctions. Since the chemical and physical properties of SnO_2_ and SnS_2_ are very similar, they can interact well at the interface. The high electron mobility through the phases promotes the adsorption of chemical species on the material’s surface, including the chemisorption of oxygen species, thereby improving the sensing properties. Along this line, some previous works have recently reported constructing SnO_2_/SnS_2_ heterojunctions for detecting toxic gas. For example, hollow core-shell-structured SnO_2_@SnS_2_ [28] and SnS_2_/SnO_2_ heterojunctions [29] both showed a superior response to NO_2_. Meanwhile, the response value of a flexible NH_3_ gas sensor based on SnO_2_/SnS_2_ is twice that of pure SnO_2_ at room temperature [30]. Inspired by the studies above, it is proposed that the construction of SnS_2_/SnO_2_ heterojunctions on SnO_2_ micro-nanostructures presents a promising strategy to overcome the bottleneck of insensitivity at room temperature. Nevertheless, to date, there has been a lack of research on the formaldehyde-sensing performances of SnS_2_/SnO_2_ composites at room temperature. Additionally, it remains a significant challenge to achieve well-defined SnS_2_/SnO_2_ composites with consistent micro-nanostructures. Furthermore, further investigation is needed to understand the sensing mechanism of SnS_2_/SnO_2_ composites, which holds more relevance in this field.

In this study, well-designed SnS_2_/SnO_2_ composites with hollow spherical structures were constructed for selective sensing of formaldehyde at room temperature, and they exhibit an ultra-high sensing response, good repeatability and stability, and ppb detection limit at room temperature. More interestingly, the SnS_2_/SnO_2_ composites have extraordinary formaldehyde selectivity, demonstrating almost no response for several common interfering gases. Density functional theory (DFT) simulations showed that the SnS_2_/SnO_2_ composites could effectively enhance the adsorption energy toward formaldehyde, contributing to its excellent selectivity. Moreover, the n–n heterojunction formed between SnO_2_ and SnS_2_ boosts the potential barrier at the interface, and the unique hollow structures bring more active sites along the material’s surface, thereby improving the sensing response. This work demonstrates that constructing SnS_2_/SnO_2_ heterojunctions on the SnO_2_ hierarchical structure efficiently develops next-generation room-temperature gas sensors with an improved sensing performance.

## 2. Experimental

### 2.1. Materials and Chemicals

The chemicals used in the experiment, including tin (IV) chloride pentahydrate (SnCl_4_·5H_2_O), sodium hydroxide (NaOH), acetic acid (C_2_H_4_O_2_), thiourea (CH_4_N_2_S), formaldehyde (HCHO), ethanol (CH_3_CH_2_OH), acetone (CH_3_COCH_3_), methanol (CH_3_OH), trimethylamine (C_3_H_9_N), benzene (C_6_H_6_), and toluene (C_7_H_8_), were analytical-grade reagents.

### 2.2. Synthesis of SnO_2_ Hollow Spheres

The SnO_2_ hollow spheres were prepared via a facile hydrothermal route. Briefly, 1.73 g SnCl_4_·5H_2_O, 1.25 g NaOH, and 30 mL of deionized water were added to a glass beaker and vigorously stirred for 30 min. Then, the clarified solution was poured into a 50 mL Teflon-lined stainless-steel autoclave and reacted at 200 °C for 24 h. After natural cooling to room temperature, the precipitate was collected, washed with deionized water and ethanol for 6 cycles, and then dried overnight at 60 °C to obtain the SnO_2_ hollow spheres.

### 2.3. Synthesis of SnS_2_/SnO_2_ Composites with Hollow Spherical Structures

Similarly, a series of SnS_2_/SnO_2_ composites with hollow spherical structures were also synthesized using a facile hydrothermal method. After introducing thiourea into the synthesis system, a portion of SnO_2_ was converted into SnS_2_; the detailed synthesis process can be found in the Appendix A.

### 2.4. Synthesis of SnS_2_/SnO_2_ Composites with Hollow Spherical Structures

The detailed characterization of the crystal structure, morphology characteristics, elemental analysis, surface valence states, specific surface area, electrochemical testing, and electron paramagnetic resonance can be found in the Appendix A.

### 2.5. Fabrication and Measurement of Sensors

The gas sensor was fabricated as follows. Firstly, a homogeneous slurry of either SnO_2_ or SnS_2_/SnO_2_ was prepared by dispersing the synthesized products (40 mg) in ethanol. This slurry was then coated onto the surface of an Al_2_O_3_ ceramic tube (4 mm in length, 1.2 mm in external diameter, and 0.8 mm in internal diameter). At each end of the Al_2_O_3_ ceramic tube, there were two Au electrodes, which were connected to two Pt lead wires. After natural drying, a Ni–Cr alloy coil was inserted through the central tunnel of the ceramic tube to act as a heater and regulate the operating temperature of the gas sensor. The Pt lead wires and the Ni–Cr heating wire were soldered on a four-corner base to fabricate the gas sensor. To enhance the stability and repeatability of the fabricated gas sensor, it was aged on a TS60 desktop (Winsen Electronics Co., Ltd., Zhengzhou, China) at 200 °C for 48 h. The gas-sensing tests were carried out in a static system (Weisheng Tech Co., Ltd., Zhengzhou, China) at room temperature. This work defines the sensor’s response as the resistance ratio, S = Ra/Rg (Rg: resistance in testing gas, Ra: resistance in the air). The schematic diagram of the synthesis process of the SnS_2_/SnO_2_ composites and the fabricated gas sensor is illustrated in Figure 1.

## 3. Results and Discussion

### 3.1. Structural and Morphological Characteristics

The appearance and morphology of the SnO_2_ and SnS_2_/SnO_2_ samples are shown in Figure 2. Obviously, pure SnO_2_ is composed of numerous spherical architectures with a diameter of approximately 2~4 μm (Figure 2a), and it can be seen that the spherical architecture is hierarchically constructed by many primary particles with a size of a few dozen nanometers in the amplifying FE−SEM images (Figure 2b), whose more detailed structure can be seen clearly in a broken SnO_2_ sphere. Interestingly, after the sulfurization of SnO_2_ (Figure 2c–h), the sample still maintains a hollow spherical structure with almost no change in size; however, compared to the SnO_2_ sample, the hollow sphere surface of SnS_2_/SnO_2_ becomes rough, which may be due to the formation of SnS_2_ nanoparticles on the SnO_2_ hollow sphere surface. The FE–SEM images exhibit porous, rough surface structures, which can absorb more gas and accelerate gas transmission, resulting in an improvement in the gas-sensing properties. According to the EDS mapping images of SnS_2_/SnO_2_−2 in Figure 2i–l, it can be seen that the three elements, namely, Sn, S, and O, are uniformly distributed on the surface, confirming the existence of SnS_2_ and SnO_2_.

Furthermore, the crystal structures of the SnO_2_ and SnS_2_/SnO_2_ samples were analyzed using XRD. As shown in Figure 3, the diffraction peaks of the SnO_2_ hollow spheres at 2θ of 26.6°, 33.9°, 38.9°, 51.8°, and 54.7° correspond to the (110), (101), (111), (211), and (220) crystal planes of SnO_2_, respectively, which is in good agreement with the tetragonal SnO_2_ phase (JCPDS No. 41–1445). After the sulfurization of SnO_2_, all the crystal phases belong to SnO_2_ and SnS_2_, confirming the formation of SnS_2_/SnO_2_ composites. Apart from the peaks belonging to pure SnO_2_, the weak diffraction peaks at about 29.3, 32.1, and 50.1° are assigned to the (101), (102), and (110) planes of SnS_2_ (JCPDS no. 21–1231). Additionally, no excess impurity peaks indicate the high purity of the as-synthesized samples.

In order to further investigate the internal microstructure and crystal structure of the SnS_2_/SnO_2_ composites, TEM and HRTEM analyses were conducted (Figure 4), which revealed that the material has a hollow spherical structure observed from the broken material (Figure 4a). Moreover, a loose and porous surface can be observed in Figure 4b, which is in accordance with the results of the FE–SEM analyses. Figure 4c–g illustrates well-defined lattice stripes of the SnS_2_/SnO_2_ composites. The crystallographic distances of 0.34, 0.18, and 0.28 nm correspond to the (110) crystallographic planes of SnO_2_, and the (110) and (102) crystallographic planes of SnS_2_, respectively. These values align with the dominant peaks observed in the XRD patterns. Furthermore, the lattice stripes observed between SnS_2_ and SnO_2_ exhibit a seamless continuity, indicating the successful formation of an n–n heterojunction.

The elemental composition and chemical state of the SnO_2_ and SnS_2_/SnO_2_−2 composites were characterized by XPS analysis, and the pure SnO_2_ comprises Sn and O (Figure 5a). In contrast, the SnS_2_/SnO_2_ composites contain Sn, O, and S. Notably, the signal peak of C is caused by contaminated carbon or corrected carbon. The Sn 3d spectrum of both pure SnO_2_ and the SnS_2_/SnO_2_ composites (Figure 5b) exhibits two peaks at 486.1 and 494.5 eV (SnO_2_) and 486.6 and 494.9 eV (SnS_2_/SnO_2_), conforming with Sn 3d_5/2_ and Sn 3d_3/2_, respectively. This confirms the presence of Sn in the samples as a Sn^4+^ form [29,31]. Additionally, the Sn 3d peaks of the SnS_2_/SnO_2_ composites exhibit a slight shift towards a higher binding energy compared to pure SnO_2_, which suggests that there is some interaction between SnO_2_ and SnS_2_ in the composites [32]. In Figure 5c, the O 1s asymmetric peak of pure SnO_2_ can be divided into three peaks at 530.0 eV, 531.0 eV, and 531.9 eV, which are attributed to lattice oxygen (O_L_), oxygen vacancies (O_V_), and surface-absorbed oxygen (O_C_), respectively [33]. The O 1s peaks of the SnS_2_/SnO_2_ composites are slightly shifted towards the high energy direction at 530.4 eV, 531.3 eV, and 532.4 eV, corresponding to O_L_, O_V_, and O_C_, respectively. Generally, the O_V_ can promote gas absorption and reaction by providing abundant active sites, and O_C_ directly affects the chemical adsorption and ionized oxygen state on the surface of sensing materials; hence, a high ratio of O_V_ and O_C_ is favorable for achieving good gas-sensing properties [34]. Through calculation, it was found that the concentration of O_V_ (25.1%) and O_C_ (11.6%) in SnS_2_/SnO_2_ are higher than those of pure SnO_2_ O_V_ (17.2%) and O_C_ (10.4%), indicating its good sensing performance. In addition, the S 2p spectrum can be divided into two peaks at 161.7 eV and 163.1 eV (Figure 5d), corresponding to S 2p_3/2_ and S 2p_1/2_, respectively [35]. The results further indicate that the as-synthesized samples are SnS_2_/SnO_2_ composites.

To further confirm the presence of oxygen vacancies in the samples, electron paramagnetic resonance (EPR) spectra of the SnO_2_ and SnS_2_/SnO_2_−2 were developed, as shown in Figure 6. The EPR spectra were measured at a temperature of 25 °C using a Bruker EMXPLUS spectrometer operating at the X band with a magnetic field modulation of 100.00 kHz. The microwave power was set to 3.170 mW and the modulation amplitude was 1.000 G [36]. Both samples revealed a strong signal corresponding to g(I) = 2.003, suggesting the presence of oxygen vacancies. The g value was determined by a comparison with a DPPH standard [37]. In addition, the signal intensity of SnS_2_/SnO_2_−2 is higher than that of SnO_2_, supporting the XPS results that the oxygen vacancy content increases after partial sulfurization of SnO_2_ to SnS_2_.

### 3.2. Gas-Sensing Properties

The formaldehyde-sensing properties of SnO_2_, SnS_2_, and SnS_2_/SnO_2_ hollow spheres were studied at room temperature. Figure 7a shows the transient resistance curve of SnO_2_, SnS_2_, SnS_2_/SnO_2_−1, SnS_2_/SnO_2_−2, and SnS_2_/SnO_2_−3 sensors to 0.1 ppm formaldehyde. Obviously, it is hard to obtain the sensing behavior of the SnO_2_ and SnS_2_ sensor at this low concentration. As for the SnS_2_/SnO_2_ sensors, an obvious response and recovery performance towards formaldehyde was observed, where the resistance decreased upon exposure to formaldehyde and recovered to its initial value after exposure to clean air, exhibiting typical n-type semiconductor characteristics. In addition, their response to formaldehyde is determined by the amount of S (Figure 7a). Notably, the SnS_2_/SnO_2_−2 sensor shows the highest response value of 1.93, meaning it has an excellent sensing ability for detecting trance formaldehyde in the atmosphere. Figure 7b shows the transient resistance curves and response value of five gas sensors to 10 ppm formaldehyde at room temperature, wherein the SnO_2_ and SnS_2_ sensor still had no response to formaldehyde at a relatively high concentration. In contrast, the SnS_2_/SnO_2_ sensor showed a significant increase in the response and recovery amplitudes, indicating that sulfurization functionalized SnO_2_ forming SnS_2_/SnO_2_ heterostructures can significantly improve the sensing ability of SnO_2_ hollow spheres. The selectivity of four sensors was investigated by comparing the sensing response to various reducing gases (formaldehyde, acetone, methanol, TMA, benzene, and toluene) with concentrations of 0.1 ppm and 10 ppm at room temperature (Figure 7c,d). Obviously, the SnO_2_ sensor had no response to any of the test gases at low or high concentrations (0.1 or 10 ppm), suggesting its poor sensing activity at room temperature. In contrast, all the SnS_2_/SnO_2_ sensors responded well to formaldehyde and were not very sensitive to other interfering gases. Specifically, the SnS_2_/SnO_2_−2 sensor was more sensitive to formaldehyde than the other sensors, demonstrating that it is more suitable for trace formaldehyde detection, and the following sensing performance was focused on the SnS_2_/SnO_2_−2 sensor.

The transient resistance curve of the SnS_2_/SnO_2_−2 sensor to a low formaldehyde concentration (0.1–1 ppm) and high formaldehyde concentration (10–100 ppm) is shown in Figure 8a,c. The corresponding response is plotted against the formaldehyde concentration, and it is apparent that the resistance decreased rapidly and reached a certain level after some time (Figure 8b,d). In contrast, upon removing formaldehyde from the chamber, the resistance gradually recovered to its original baseline value, confirming its good reproducibility. As the formaldehyde concentration increased, the response improved significantly (0.1–1 ppm), but when the formaldehyde concentration was above 10 ppm, the magnitude of the rise in response gradually slowed down due to the high surface coverage of formaldehyde [38]. In addition, the response versus the formaldehyde concentration (0.1–1 ppm) presents a good linear relationship with a fitting slope of 8.44408 ppm^−1^ (Figure 8b) [39]. The theoretical detection limit is calculated to be approximately 5.81 ppb based on the signal-to-noise ratio (see the Appendix A for details). Such results suggest that the SnS_2_/SnO_2_−2 sensor in this work can potentially monitor ppb-level formaldehyde in the environment.

Repeatability and stability are essential parameters for practical applications. The repeatability of the SnS_2_/SnO_2_−2 sensor was investigated by exposing it to 0.1 ppm or 10 ppm formaldehyde in five cycles of response-recovery at room temperature (Figure 8e). Each cycle exhibited similar adsorption-desorption trends, and the initial resistance value did not change significantly. Additionally, the response also showed slight fluctuations in each trial, implying its good repeatability. Furthermore, the formaldehyde-sensing response of the SnS_2_/SnO_2_−2 sensor was monitored for 35 days to identify its long-term stability (Figure 8f). Interestingly, there was no significant decrease in the response over the testing period; even 35 days later, the response remained at 96.89% (0.1 ppm) and 97.94% (10 ppm), indicating its good long-term stability. At room temperature, humidity is also an essential factor affecting the sensor’s practical application. The effect of relative humidity (RH) on the formaldehyde-sensing response (0.1 ppm or 10 ppm) at room temperature was explored, and the result is shown in Figure 8g. It was found that in the range of RH from 25% to 55%, the response to 0.1 ppm or 10 ppm formaldehyde decreased slightly with increasing RH levels. In contrast, the decrease in response was relatively higher when the RH value was above 55%, which could be originated from the adsorption competition between oxygen species and water molecules on the sensor surface, hindering the redox reaction of the target gas [40]. A reduction of the baseline resistance as the RH levels increased was also observed (Figure 8h), implying the occurrence of surface reactions between water molecules and the adsorbed oxygen species [41]. In addition, Table 1 compares the results of this work and others reported in the literature for formaldehyde detection. Compared to the previously described sensing material, the SnS_2_/SnO_2_−2 composites offer more significant potential for a formaldehyde-sensing ability at room temperature due to their high responses and low detection limit.

### 3.3. Gas-Sensing Mechanism

Generally, the sensing mechanism of SnO_2_ sensors is related to the resistance modulation caused by the chemical interaction between the absorbed target gas and ionized oxygen species on the SnO_2_ [51]. In this work, the SnO_2_ and SnS_2_/SnO_2_ composites showed a typical n-type sensing feature, meaning that electrons act as the major charge carriers for sensing reactions. This is also confirmed by the Mott–Schottky analyses, and both SnO_2_ and SnS_2_/SnO_2_−2 hollow spheres show a positive slope in the M–S plots (Figure 9a), indicating that electrons act as the leading carriers and exhibit n-type semiconductor conduction characteristics. Subsequently, the flat band potential (U_FB_) of the composite was ascertained by calculating the intercept of the linear segment of the Mott–Schottky curve of the SnS_2_/SnO_2_−2 composite with the potential axis [52]. The U_FB_ of the composite is −0.43 V (standard hydrogen electrode, SHE). For the pure SnO_2_ hollow spheres, electrons from the conduction band can react with the oxygen molecules in the ambient atmosphere to form oxygen anions (O_2_^−^, O^−^, and O^2−^), depending on the operational temperatures. This process forms an electron depletion layer on the SnO_2_ surface, thereby increasing the resistance (Ra) of the SnO_2_ sensor in the air. Usually, at room temperature, the surface oxygen species were mainly O_2_^−^. Upon exposure to reducing gas (formaldehyde), the ionized oxygen species could react with the formaldehyde molecules to release the trapped electrons into the SnO_2_ conduction band, thereby narrowing the thickness of the electron depletion layer and reducing the resistance value of the sensor (Rg). The chemical reactions are as follows [53,54,55]:(1)$$O_{2}(gas)\to O_{2}(ads)$$
(2)$$O_{2}\left(ads\right)+e^{-}\to O_{2}^{-}(ads)$$
(3)$$HCHO(ads)+O_{2}^{-}(ads)\to {CO}_{2}(gas)+H_{2}O+e^{-}$$

The reasons for the improved gas-sensing performance of the SnS_2_/SnO_2_ composite materials, unlike single-type SnO_2_, are as follows.

(1)The existence of an n–n heterojunction plays a crucial role in enhancing the sensing performance. The difference between the electronic work function of SnO_2_ and SnS_2_ makes the electrons in SnS_2_ flow to SnO_2_ until the Fermi level reaches equilibrium [28] when they contact each other (Figure 9b). The transfer of electrons and the significant difference in electron work functions result in band bending within the material, further causing the accumulation of electrons at the surface of SnO_2_ and the depletion of electrons at the surface of SnS_2_. Meanwhile, a potential barrier is generated between the heterojunction architecture. As the SnS_2_/SnO_2_ is exposed to the air environment, more ionized oxygen species are absorbed on the surface of the sensing materials, leading to a more significant initial resistance state immediately. When it is exposed to formaldehyde, the sensing reaction of the oxygen species with formaldehyde releases more electrons back to the conduction band. This process narrows the electron depletion layer, dramatically decreasing the sensor resistance with the reduced heterojunction potential barrier height; thus, the SnS_2_/SnO_2_ heterojunction configuration significantly enhances the sensor’s capabilities.(2)The oxygen/formaldehyde adsorption capacity has an important impact on the gas-sensing performance of the materials. The XPS and EPR analyses show that the SnS_2_/SnO_2_ composites process more oxygen vacancies, implying their high oxygen adsorption capacity. Upon exposure to formaldehyde, more oxygen species means more formaldehyde molecules can react, consequently leading to a high gas-sensing ability. At the same time, the enhanced adsorption ability of the SnS_2_/SnO_2_ composites was revealed through DFT calculations, which were performed employing the CASTEP module in Materials Studio software (see the Supporting Information and Appendix A for details). As shown in Figure 9c, when the sensor is in contact with the tested gas molecules, the adsorption energy of formaldehyde on the (110)/(101) plane of SnS_2_/SnO_2_ is −1.11 eV, which is much larger than that of the other tested gases (acetone: −0.53 eV, methanol: −0.25 eV, toluene: −0.71 eV, benzene: −0.53 eV, TMA: −0.88 eV, formaldehyde: −1.16 eV). This indicates a strong interaction between formaldehyde and the SnS_2_/SnO_2_ surface, directly proving the improved sensing performance to formaldehyde from the energy point of view.(3)The unique structural merits, including the hollow and porous structure, are also essential in improving the gas-sensing performance. The hollow, mesoporous structure of the SnS_2_/SnO_2_ composites significantly contributes to the specific surface area (Figure 9d). The BET measurements (see the Supporting Information and Appendix A for details) revealed that both SnO_2_ and SnS_2_/SnO_2_ hollow spheres possess a high specific surface area, and the SnS_2_/SnO_2_-2 hollow spheres (92.5 m^2^ g^−1^) have a higher specific surface area than that of SnO_2_ (87.4 m^2^ g^−1^). This indicates that the SnS_2_/SnO_2_−2 hollow spheres can provide many active sites for the adsorption of oxygen species and formaldehyde gas, booting the resistance modulation. In addition, the porous channel structure can essentially promote the penetration efficiency of air/target molecules in the sensing interaction, boosting the reaction of formaldehyde and oxygen species.

Overall, the synergistic effect of the n–n heterojunctions between SnO_2_ and SnS_2_, the strong oxygen/formaldehyde adsorption capacity, and the unique structural merits, including the hollow and porous structure, enhance the formaldehyde-sensing performance in the SnS_2_/SnO_2_ composites; however, excessive SnS_2_ can decrease the sensor response, as it can hide the SnO_2_ surface and become the dominant conductive path, hindering the desired sensing function. Moreover, increasing the SnS_2_ content further degrades the quality of the n–n heterojunction, weakening its effect. Based on the results, the SnS_2_/SnO_2_−2 sensor with optimized decoration significantly improves the formaldehyde-sensing properties of the material, making it a more effective gas sensor for detecting this harmful gas.

## 4. Conclusions

In summary, SnS_2_/SnO_2_ heterostructures with hollow spherical structures were synthesized by a simple hydrothermal method for highly sensitive and selective formaldehyde detection. Benefiting from the synergistic effect of the n–n heterojunction between SnO_2_ and SnS_2_, strong oxygen/formaldehyde adsorption capacity, and unique structural features, including the hollow and porous structure, the SnS_2_/SnO_2_ sensor presents a superior formaldehyde-sensing performance at room temperature. Remarkably, the SnS_2_/SnO_2_−2 sensor demonstrates exceptional selectivity, detecting formaldehyde in concentrations as low as 0.1 ppm, with the highest sensing response of 1.93. Furthermore, the sensor exhibits good repeatability, long-term stability, and an outstanding theoretical detection limit. To gain further insight into the enhanced gas-sensing performance, we conducted DFT calculations, which shed light on the underlying mechanism. The excellent comprehensive gas-sensing properties enable SnS_2_/SnO_2_ hollow spheres to position themselves as highly selective gas-sensing platforms for the detection of trace amounts of formaldehyde. This technology is well-suited for energy-saving and portable detection systems, meeting the demands of various applications.

## Figures and Tables

**Figure 1 nanomaterials-13-02493-f001:**
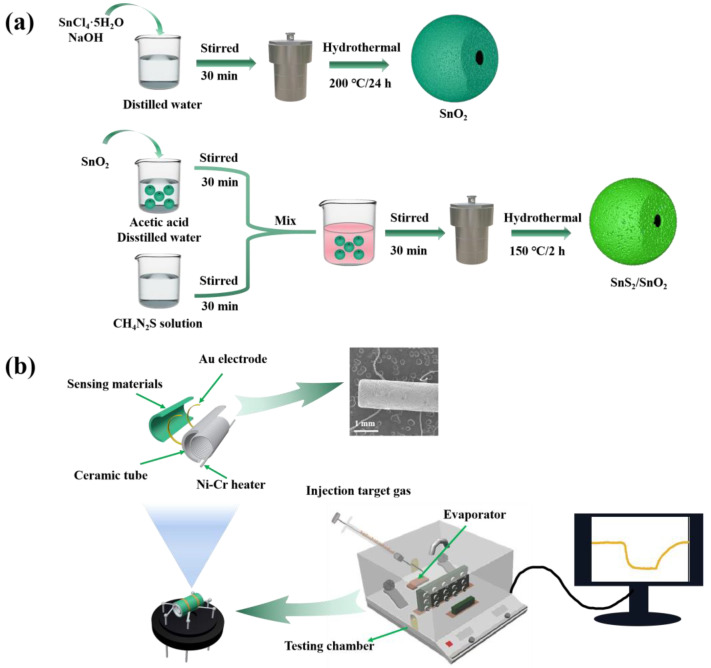
(**a**) Schematic illustration of the synthesis process of the SnS_2_/SnO_2_ composites. (**b**) Schematic illustration and corresponding SEM image of the gas sensor, and schematic illustration of measurement system.

**Figure 2 nanomaterials-13-02493-f002:**
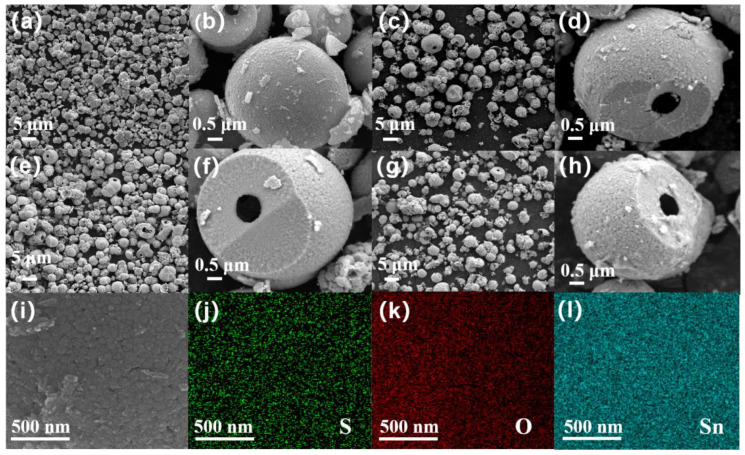
The FE–SEM images of (**a**,**b**) SnO_2_, (**c**,**d**) SnS_2_/SnO_2_−1, (**e**,**f**) SnS_2_/SnO_2_−2, and (**g**,**h**) SnS_2_/SnO_2_−3. (**i**–**l**) EDS elemental mapping images of SnS_2_/SnO_2_−2.

**Figure 3 nanomaterials-13-02493-f003:**
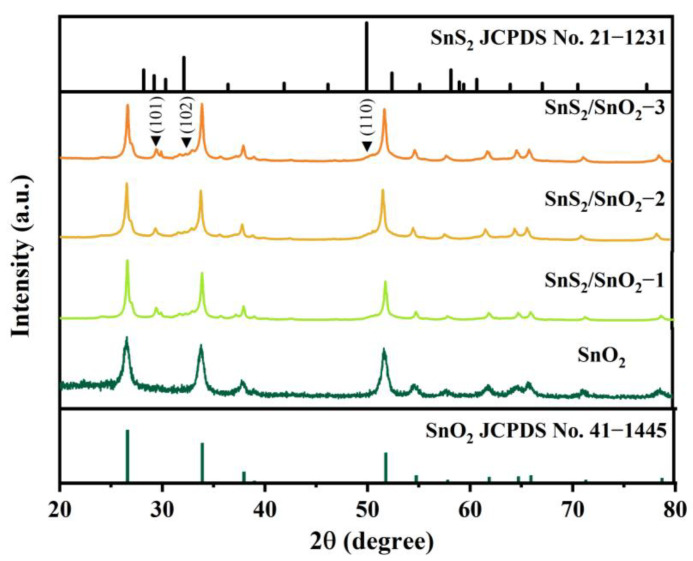
XRD patterns of SnO_2_ and SnS_2_/SnO_2_ samples.

**Figure 4 nanomaterials-13-02493-f004:**
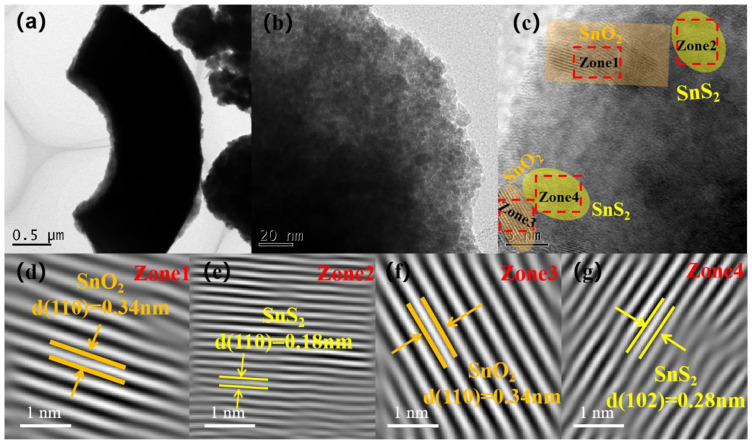
TEM image (**a**) and (**b**–**g**) corresponding HR–TEM image of SnS_2_/SnO_2_−2 composites.

**Figure 5 nanomaterials-13-02493-f005:**
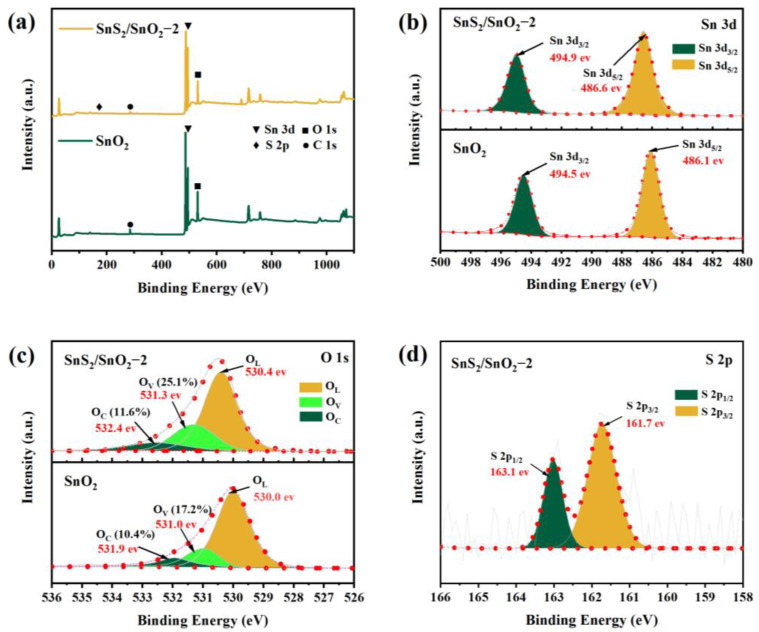
XPS spectra of SnO_2_ and SnS_2_/SnO_2_−2 samples. (**a**) Survey spectrum, (**b**) Sn 3d spectrum, (**c**) O 1s spectrum, and (**d**) S 2p spectrum.

**Figure 6 nanomaterials-13-02493-f006:**
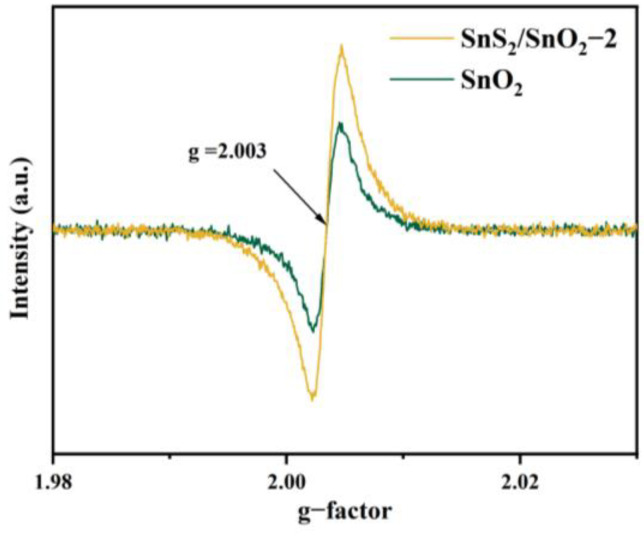
EPR spectra of SnO_2_ and SnS_2_/SnO_2_−2 hollow spheres.

**Figure 7 nanomaterials-13-02493-f007:**
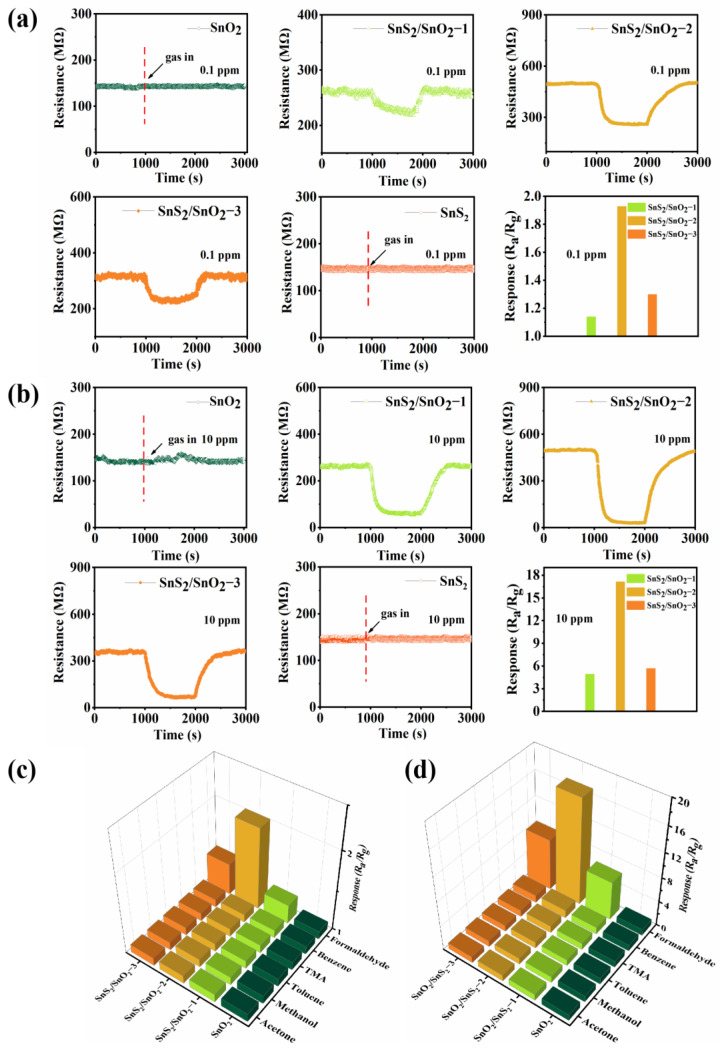
Gas-sensing properties of the sensors at room temperature: dynamic response recovery curve and corresponding response of SnO_2_, SnS_2_, and SnS_2_/SnO_2_ hollow spheres at room temperature to (**a**) 0.1 ppm formaldehyde and (**b**) 10 ppm formaldehyde; selectivity of SnO_2_ and SnS_2_/SnO_2_ sensors to six different gases at room temperature at (**c**) 0.1 ppm and (**d**) 10 ppm.

**Figure 8 nanomaterials-13-02493-f008:**
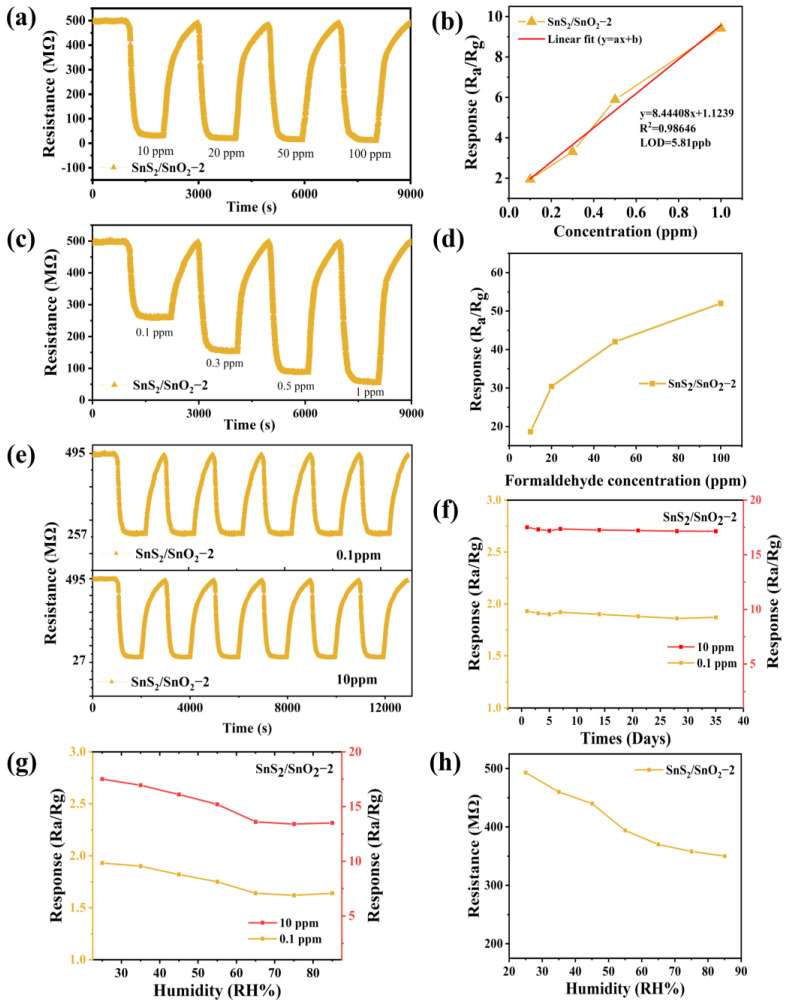
Gas-sensing properties of the SnS_2_/SnO_2_−2 sensor at room temperature: (**a**) dynamic response recovery curve and (**b**) correlation curve of the response with formaldehyde concentration (0.1–1 ppm); (**c**) dynamic response recovery curve and (**d**) corresponding response with formaldehyde concentration (10–100 ppm); (**e**) repeatability, (**f**) long-term stability, (**g**) response changes under varying humidity to formaldehyde (0.1 ppm or 10 ppm), and (**h**) the variation of baseline resistance versus relative humidity.

**Figure 9 nanomaterials-13-02493-f009:**
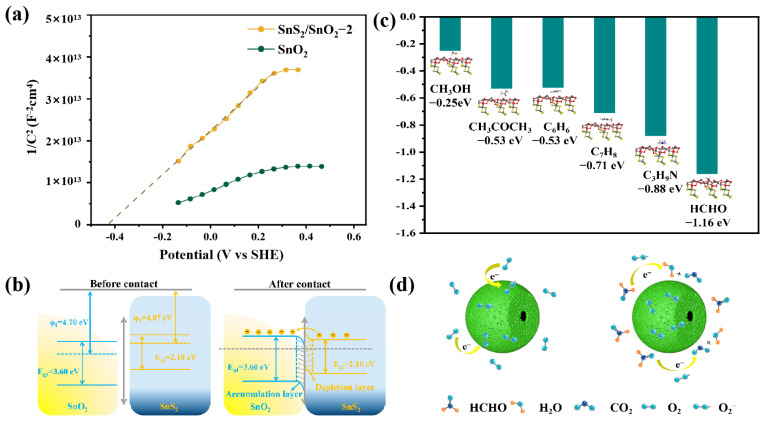
(**a**) Mott–Schottky plots of the SnO_2_ and SnS_2_/SnO_2_−2 hollow spheres; (**b**) energy band structure diagram of the SnS_2_/SnO_2_ heterojunctions; (**c**) DFT-calculated adsorption energies of the SnS_2_/SnO_2_ heterojunction surface; (**d**) schematic diagram of sensing mechanism of SnS_2_/SnO_2_ hollow spheres in air and formaldehyde.

**Table 1 nanomaterials-13-02493-t001:** Comparison of sensing ability of gas sensors based on different sensing materials toward formaldehyde.

Materials	Temperature (°C)	Concentration (ppm)	Response (Ra/Rg)	Res./Rec. Time (s)	LOD	References
Sn_3_O_4_/rGO	150	100	44	4/125	1 ppm	[1]
PdPt/SnO_2_	190	1	83.7	5/7	50 ppb	[42]
In_2_O_3_/TiO_2_	RT	1	3.8	28/50	0.06 ppm	[43]
In_2_O_3_/ANS/rGO	RT	0.5	2.4	119/179	5 ppb	[44]
Ni-In_2_O_3_/WS_2_	RT	20	32	76/123	15 ppb	[45]
C/rh-In_2_O_3_	120	50	330	12/355	11 ppb	[46]
MXene/Co_3_O_4_	RT	10	9.2	0.17/0.19	0.01 ppm	[47]
Bi doped Zn_2_SnO_4_/SnO_2_	180	50	23.2	16/9	--	[48]
2 at% Al-doped ZnO	320	50	6.8	81/21	0.5 ppm	[49]
SnO_2_/ZSM−5	250	10	11.67	37/115	2 ppm	[50]
SnS_2_/SnO_2_	RT	0.1	1.93	227/424	5.81 ppb	This work

## Data Availability

Not applicable.

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
