# Peer review of "In Situ Fabrication of SnS2/SnO2 Heterostructures for Boosting Formaldehyde−Sensing Properties at Room Temperature"

_nanomaterials, 2023, doi:10.3390/nano13172493_

Round 1

Reviewer 1 Report

XRD patterns SnS2/SnO2 samples: The data presented does not really explain the composition of SnS2 and SnO2 in the mixture. It is also not clear whether the two different particles are mixtures doped or in solid solution. This needs to be carefully anlyzed. From the present description readers would not know the structural properties of the materials. Authors are advised to look into this seriously. The XRD signal e.g. (100) is not a characteristic signal of SnS2 based on the JCPDS data (23-0677). Please revisit the data with better evaluation 

TEM image (a) and (b)-(c) corresponding HR-TEM image of SnS2/SnO2-2: The lattice distance evaluation of SnS2 is not clear. Authors are advised to re-image high crystalline primary particles. The data presented in the manuscript misleads the readers.   

XPS spectra of SnO2 and SnS2/SnO2-2 samples: This reviewer is not sure why this measurement was performed. How are  Sn 3d3/2 and Sn 3d5/2 signals differing from SnS2 and SnO2? Again the message of this measurement is not clear.

EPR spectra of SnO2 and SnS2/SnO2-2: The EPR spectrum presented inthe manuscript does not show evidence of oxygen vaccancies. What was the standard used for the measurement (see Appl. Magn. Reson., 1994, 6, 341-345)? Based on the magnetic moment of the reference, the samples need to be characterized. In this manuscript all these explanation are lacking. Based on the reference 29, which shows the similar EPR with reduced intensity (due to sintering), authors assume that this should be the same. To find such vaccacies, detail crystallographic measurements has to be performed (pXRD with high intensity, Reitveld refinements, analyzing how the cell parameters are varying with the stocichiometric particles).

Gas sensing measurements: (1) The sensor fabrical procedure is not available int he mansucript. Authors are advised to describe this in the main text (2) Why SnS2/SnO2-2 is not really clear, why should this sensor differ from the other ones? (3) the sensor characterization is also not fully performed. Please see the reference: Sens. and Actuators B, 2012, 161(1), 740-747.

Why does the conduction and valence band bend in opposite direction for the same n-type SnO2 or SnS2 in the mixture? What is the flat band potential of this material? When does the Fermi energy equilibrate?  

XRD patterns SnS2/SnO2 samples: The data presented does not really explain the composition of SnS2 and SnO2 in the mixture. It is also not clear whether the two different particles are mixtures doped or in solid solution. This needs to be carefully anlyzed. From the present description readers would not know the structural properties of the materials. Authors are advised to look into this seriously. The XRD signal e.g. (100) is not a characteristic signal of SnS2 based on the JCPDS data (23-0677). Please revisit the data with better evaluation 

TEM image (a) and (b)-(c) corresponding HR-TEM image of SnS2/SnO2-2: The lattice distance evaluation of SnS2 is not clear. Authors are advised to re-image high crystalline primary particles. The data presented in the manuscript misleads the readers.   

XPS spectra of SnO2 and SnS2/SnO2-2 samples: This reviewer is not sure why this measurement was performed. How are  Sn 3d3/2 and Sn 3d5/2 signals differing from SnS2 and SnO2? Again the message of this measurement is not clear.

EPR spectra of SnO2 and SnS2/SnO2-2: The EPR spectrum presented inthe manuscript does not show evidence of oxygen vaccancies. What was the standard used for the measurement (see Appl. Magn. Reson., 1994, 6, 341-345)? Based on the magnetic moment of the reference, the samples need to be characterized. In this manuscript all these explanation are lacking. Based on the reference 29, which shows the similar EPR with reduced intensity (due to sintering), authors assume that this should be the same. To find such vaccacies, detail crystallographic measurements has to be performed (pXRD with high intensity, Reitveld refinements, analyzing how the cell parameters are varying with the stocichiometric particles).

Gas sensing measurements: (1) The sensor fabrical procedure is not available int he mansucript. Authors are advised to describe this in the main text (2) Why SnS2/SnO2-2 is not really clear, why should this sensor differ from the other ones? (3) the sensor characterization is also not fully performed. Please see the reference: Sens. and Actuators B, 2012, 161(1), 740-747.

Why does the conduction and valence band bend in opposite direction for the same n-type SnO2 or SnS2 in the mixture? What is the flat band potential of this material? When does the Fermi energy equilibrate?  

Reviewer 2 Report

In this article, D. Meng et.al, demonstrated the synthesis of SnS2/SnO2-based heterostructure nanocomposite for enhanced ethanol sensing. The design and organization of the manuscript are well and satisfactory, and the prosed mechanism and results are good in consistent with the DTF studies as well as published literature. However, there are some issues that authors should focus on to enhance the quality of the manuscript, as noted below before it can be published.

1)      What is the novelty of structure? There are many reports which already has published based on SnS2/SnO2 composite heterostructure. And how your structure is much more sensitive to formaldehyde gas. Please provide these missing details in the introduction to further enhance the novelty of your study.

2)      Please check English writing. There are some minor errors in the sentence formations.

3)      Please supplement the elemental percentage of SnS2/SnO2 heterostructure and their pristine counter parts.

4)      There are no other characterization techniques to support the SnS2 presence. Please provide and compare their electrical, structural, chemical and gas sensing characterizations: XRD, XPS.

5)      From the XRD, have the authors noticed any peak shift, and why did the FWHM of the peaks become smaller? Please explain the crystalline nature of composite materials in the XRD section.

6)      From XPS O 1s spectra, authors have discussed the OV and OC and their significance in gas sensing. How about the OL? Please explain its role.

7)      Provide dynamic gas sensing response curve of pure SnS2.

8)      What are the possible reasons for the SnS2/SnO2-2 sample has superior sensing properties? Explain more in detail in the revised manuscript.

9)      In Figure 8b, the plots were in between only two points. Is it ok to estimate the linearity with low error? My opinion is it could be better if the plotting would be more than 2 points. Please consider.

10)  What is the resistance level of the sensor under various relative humidity?

11)  How about long-term stability?

12) On page NO: 10, line NO: 282, oxygen anions were replaced s (O2- , O- , and O2- ).please modify.

13)  What is the thickness of the as-prepared sensor sample? Does the thickness is uniform for all the sensors (pristine and composite). Please provide the details in the revised manuscript because thickness plays a key role.

14)  Please improve the quality of DFT studies in Figure 9c for better understand by enlarging the image.

15)  For the theoretical detection limit, there are particular supporting references. Please add.

16)  This work investigated the SnS2/SnO2 based nanocomposite for formaldehyde sensing. Some relative papers may enrich the concepts and background of this work as references: Sensors and Actuators B: Chemical 390 (2023): 133976 Nano Res. 16, 7682–7695 (2023): ACS Applied Materials & Interfaces 12.51 (2020): 57207-57217, J. Hazard. Mater., 427 (2022), p. 128174, Sensors and Actuators B: Chemical 359 (2022): 131612.

Minor grammatical errors in the sentence formations. This can be resolved if the author can do a quick proof read.

Round 2

Reviewer 2 Report

Authors have revised the manuscript well, and have answerd all my concerns. Hence this manuscipt can be accepted for the publication.